CERN-TH-2023-238, Nikhef 2023-026, OUTP-23-16P

# Introduction to the PanScales framework, version 0.1

Melissa van Beekveld[1], Mrinal Dasgupta[2], Basem Kamal El-Menoufi[2,3], Silvia Ferrario Ravasio[4], Keith Hamilton[5], Jack Helliwell[6], Alexander Karlberg[4], Rok Medves[6], Pier Francesco Monni[4], Gavin P. Salam[6,7], Ludovic Scyboz[3,6], Alba Soto-Ontoso[4], Gregory Soyez[8], Rob Verheyen[5]

**1** Nikhef, Theory Group, Science Park 105, 1098 XG, Amsterdam, The Netherlands
**2** Department of Physics & Astronomy, University of Manchester, Manchester M13 9PL, United Kingdom
**3** School of Physics and Astronomy, Monash University, Wellington Rd, Clayton VIC-3800, Australia
**4** CERN, Theoretical Physics Department, CH-1211 Geneva 23, Switzerland
**5** Department of Physics and Astronomy, University College London, London, WC1E 6BT, UK
**6** Rudolf Peierls Centre for Theoretical Physics, Clarendon Laboratory, Parks Road, University of Oxford, Oxford OX1 3PU, UK
**7** All Souls College, Oxford OX1 4AL, UK
**8** Université Paris-Saclay, CNRS, CEA, Institut de physique théorique, 91191, Gif-sur-Yvette, France

mbeekvel@nikhef.nl, mrinal.dasgupta@manchester.ac.uk, basem.el-menoufi@monash.edu, silvia.ferrario.ravasio@cern.ch, keith.hamilton@ucl.ac.uk, jack.helliwell@physics.ox.ac.uk, alexander.karlberg@cern.ch, pier.monni@cern.ch, gavin.salam@physics.ox.ac.uk, ludovic.scyboz@monash.edu, alba.soto.ontoso@cern.ch, gregory.soyez@ipht.fr

## Abstract

**In this article, we document version 0.1 of the PanScales code for parton shower simulations. With the help of a few examples, we discuss basic usage of the code, including tests of logarithmic accuracy of parton showers. We expose some of the numerical techniques underlying the logarithmic tests and include a description of how users can implement their own showers within the framework. Some of the simpler logarithmic tests can be performed in a few minutes on a modern laptop. As an early step towards phenomenology, we also outline some aspects of a preliminary interface to Pythia8.3, for access to its hard matrix elements and its hadronisation modules.**

**The code is available from https://gitlab.com/panscales/panscales-0.X**

# 1  Introduction

Parton showers lie at the core of the majority of experimental and phenomenological studies in collider physics. At the LHC, they connect the electroweak and TeV momentum scales of hard-scattering processes, where the relevant degrees of freedom are perturbative quarks and gluons, with the non-perturbative physics of hadrons at scales of a few hundred MeV. As such, parton showers account for physics across several orders of magnitude in momentum scales. In QCD, large logarithms typically appear in the presence of large momentum scale hierarchies, which have to be resummed to all orders in the strong coupling to obtain physically sensible results. One of the frontiers of the development of parton showers is to understand, demonstrate and improve their logarithmic accuracy, with analytic resummations providing crucial inputs, as well as reference results for comparison.

This paper documents the first public release of a new parton showering code, PanScales, version 0.1. It has been developed as part of a series of articles [1–9] investigating how to design parton shower algorithms that provide controlled and verifiable logarithmic accuracy, together with parallel analytical work on approaches to resummation at higher logarithmic accuracy and their connection with parton showers [10–17]. Several other groups have also recently been working on the question of logarithmic accuracy in showers, see e.g. Refs. [18–25].

This PanScales release includes two main NLL-accurate parton showers, PanGlobal

and PanLocal. They have had their next-to-leading-logarithmic (NLL) accuracy tested for $e^+e^-$ [1] and colour-singlet production in $pp$ [5,6] collisions, as well as Deep Inelastic Scattering (DIS) and Vector Boson Fusion (VBF) processes [8]. They include state-of-the-art handling of subleading colour corrections, which for many processes and observables allows for full colour accuracy at LL (and often beyond) [2,5]. They also include the treatment of both collinear and soft spin correlations [3–5], and first steps towards matching [7], as well as elements towards NNLL accuracy [9] (the latter two just for $e^+e^-$ collisions). Finally, the codebase contains early versions of features that are yet to be discussed in physics research papers, in particular an interface with the PYTHIA8.3 event generator [26], which can be used to provide hard-process generation and hadronisation. Despite the inclusion of the interface to PYTHIA8.3, the code is not yet at a stage of maturity that is suitable for extensive comparisons to experimental data. This is notably because of the absence of finite quark-mass effects, the need for further work on matching with higher (fixed) order effects, as well as tuning of the non-perturbative parameters of the shower and of any hadronisation model with which it is used.

This manuscript is structured as follows. Section 2 focuses on basic usage of the code, illustrating: the build procedure (section 2.1); stand-alone event generation (section 2.2); the use of the code for carrying out basic logarithmic tests of parton showers (section 2.3); usage with a preliminary interface to PYTHIA8.3 (section 2.4); and details for carrying out validation of the code and building it with higher numerical precision (section 2.5). Section 3 illustrates some of the techniques that underlie the logarithmic tests, discussing both double-logarithmic global event shape observables (section 3.1) and single-logarithmic non-global observables (section 3.2). Section 4 gives a brief discussion of how to use the PanScales framework to implement a new shower, which provides a relatively straightforward way to gain access to the colour, spin-handling and logarithmic-accuracy testing facilities. We close in section 5 with an outlook.

## 2 Basic usage

The PanScales code requires a C++14 compiler and a Fortran 95 compiler, the GSL library, CMake ($\geq$ 3.7) and, for some scripts, Python ($\geq$ 3.6) with matplotlib installed. Some features (higher-precision builds) require the MPFR [27] and QD [28] libraries (see section 2.5 for details). It includes several third party codes, notably fjcore [29] for jet finding and hoppet [30] for PDF handling and the Catch2 library for unit-testing (see the 3rdPartyCode.md file for further details). The code can also be linked to LHAPDF [31] and to PYTHIA8.3. The PanScales code is released under the GNU GPL v3 license.

### 2.1 Downloading and building the code

PanScales can be obtained from the git repository

```
git clone --recursive https://gitlab.com/panscales/panscales-0.X
```

The main code is in the shower-code/ subdirectory. To build the code and examples in double-precision, do the following

```
cd panscales-0.X/shower-code
../scripts/build.py -j
```

The scripts/build.py script uses CMake to organise the build, which by default is placed in the build-double/ subdirectory. Advanced use of CMake is described in the BUILD.md file.

## 2.2 Standalone event generation

To run showering for the $e^+e^- \to q\bar{q}$ process and analyse the events, do

```
build-double/example-ee -shower panglobal -beta 0 -process ee2qq \
        -physical-coupling -rts 91.1876 -nev 100000 \
        -out example-results/example-ee.dat
```

This will take about $5-10$ seconds, and produce an output file with histograms for a range of event shapes. Runs are generally configured with command-line options, for example with the first line indicating the use of the PanGlobal shower. The shower ordering variable is $v = k_t e^{-\beta_{\mathrm{PS}}|\eta|}$, with $k_t$ and $\eta = -\ln\tan\theta/2$ respectively the transverse momentum and pseudorapidity of the emission with respect to its parent. The choice $\beta_{\mathrm{PS}} = 0$ (set with the `-beta 0` command-line argument) corresponds to transverse-momentum ordering.

The available command-line options can be explored with the `-h` flag, and command-line options can also be placed in a card-file and read with the `-argfile card-file.txt` option. For convenience, the main options are also listed in an `OPTIONS.md` file (any executable can be made to generate such a file by adding `-markdown-help` to the command line). More details on the PanScales interface can be found by examining the `example-ee.cc` file. Much of the code also has `doxygen` documentation, which can be obtained by running `doxygen` from the `shower-code/` directory.

Note that in standalone mode, as given above, currently all events have unit weight, i.e. in order to recover a physical cross-section from the above run the histograms should be multiplied by the appropriate cross section for the hard process. Furthermore, in most cases the Born event is a fixed configuration rather than being sampled over.

A final comment is that the event record (in the class `panscales::Event`) holds only the particles as they appear after all showering, rather than containing all intermediate steps as in some other codes. The reason for this is that for showers with global recoil, storing all intermediate steps would take memory of order $n^2$ for an $n$-particle event. By default the code prints the first event, but this can be changed to print $N$ events with the `-max-print` $N$ option.

Similar examples for $pp \to Z$ and DIS are given respectively in the `example-pp.cc` and `example-dis.cc`, with further explanations in an `EXAMPLES.md` file. Note that the above examples do not by default integrate over the hard-process kinematics. That functionality is instead available through the interface with Pythia8.3 (cf. section 2.4).

## 2.3 Logarithmic tests

Within the `shower-code/` directory, the `example-global-nll-ee.py` script, and associated `example-global-nll-ee.cc` program, illustrate how to carry out a basic test of the NLL accuracy of a shower for global event-shape observables. It can be used as

```
./example-global-nll-ee.py --njobs NJOBS --shower panglobal
```

and will test the NLL accuracy of the $e^+e^-$ PanGlobal shower with $\beta_{\mathrm{PS}} = 0$ (i.e. $k_t$ ordered), for event shapes like the Cambridge-algorithm $y_{23}$ [32] and Lund observables [1]. The Lund observables measure either the sum or the maximum of the $k_{ti}e^{-\beta_{\mathrm{obs}}|\eta_i|}$ for primary Lund-plane declusterings $i$ [33]. The NLL test works with $\beta_{\mathrm{obs}} = 0$, i.e. examining just the relative transverse momenta ($k_t$) of the declusterings.

The `--njobs` flag takes an integer which should be equal to the number of cores that one wishes to run on. Depending on the machine, the script takes around 15 to 50 CPU minutes, i.e. a few minutes of wall-time on a modern multi-core machine. On completion, for each observable, it will produce plots such as those in Fig. 1, showing the $\alpha_s \to 0$ limit

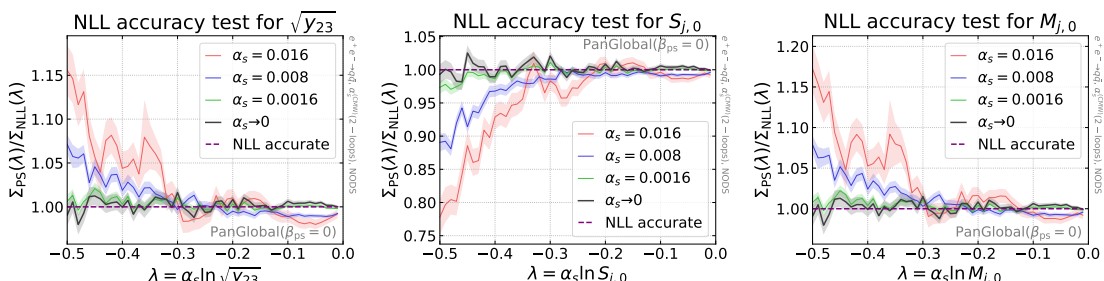

Figure 1: Example results from tests of NLL accuracy of the PanGlobal ($\beta_{\mathrm{PS}} = 0$) shower for three global event-shape observables, the Cambridge algorithm $y_{23}$ and the Lund $\beta_{\mathrm{obs}} = 0$ sum and maximum observables, corresponding respectively to the three panels. Each panel shows the ratio of the shower cumulative-distribution result to the NLL calculation. Each coloured line corresponds to a specific value of $\alpha_s$, while the black line gives the extrapolation to $\alpha_s = 0$. The results are shown as a function of the maximum allowed value of $\lambda = \alpha_s \ln O$, where $O$ is the observable.

of the ratio $\Sigma_{\mathrm{shower}}(\lambda)/\Sigma_{\mathrm{NLL}}(\lambda)$, where $\Sigma(\lambda)$ is the cross section for $\alpha_s \ln O$ to be smaller than $\lambda$, with $O$ being the value of the observable. The plot is given as a function of $\lambda$, as is standard for NLL tests [1]: for an NLL-correct shower, the $\alpha_s \to 0$ limit of the ratio will be equal to 1. The user can examine the results for a non-NLL shower by replacing `panglobal` on the command line with `dipole-kt`, which provides a standard $k_t$-ordered dipole shower [5], much like those [34–36] in standard public tools.

The `shower-code` directory also includes an `example-nonglobal-nll-ee.py` script for testing non-global logarithms in the context of energy flow into an angular slice. It can be used as follows

```
./example-nonglobal-nll-ee.py --njobs NJOBS --shower panglobal
```

The script again generates a plot with the ratio to the NLL (single-logarithmic) result.

The above command lines serve mainly to illustrate the more straightforward logarithmic accuracy tests and provide only a subset of the functionality required for a full set of tests. For example, the global event-shape tests in the `example-global-nll-ee.py` script are limited to $\beta_{\mathrm{PS}} = \beta_{\mathrm{obs}} = 0$. Further discussion of considerations for logarithmic tests is given in section 3.

## 2.4 Usage with Pythia8

While we do not yet recommend the PANSCALES code for phenomenological production purposes, those wishing to start exploring such applications may try the interface with the PYTHIA8.3 [26] generator code, which we have tested with version 8.3.10. This enables the use of PYTHIA8.3 to generate the hard process,[1] as well as for hadronisation, while PANSCALES carries out the parton showering at the scales in between. It also enables access to PYTHIA8.3 I/O, e.g. for outputting HepMC files [37].

To compile and run the code, enter the `pythia-interface/` directory and run[2]

---

[1] So far only a limited set of processes is supported in the interface, e.g. because of the setup of subleading-colour information for the hard process, which is currently handled manually.

[2] Users who wish to reuse an existing PYTHIA8.3 installation should see `pythia-interface/README.md` for instructions. For users who already have a version of PYTHIA8.3 installed but also run the `get-pythia.sh` script, care should to be taken about conflicts between the two versions of PYTHIA8.3 (e.g. library paths for dynamic linking leading to inconsistent versions).

```
./get-pythia.sh
../scripts/build.py --build-lib -j --with-lhapdf
build-double/main-dy -physical-coupling -lhapdf-set CT14lo \
      -shower panglobal -nev 1e6 -out main-dy.dat
```

This will simulate Drell-Yan production and histogram the $Z$ rapidity, mass and transverse momentum. It does not (yet) include matching to higher-order matrix elements, so kinematic distributions such as the transverse momentum of the colour singlet $p_{tZ}$ are sensible only in the resummation region, i.e. at low $p_{tZ}$. This example uses the CT14lo set [38] from LHAPDF [31],[3] and will produce warnings concerning $x$ regions where the PDF set is badly behaved. Most other PDF sets have issues with negative or zero parton distribution functions, especially at large $x$, that cause the PanScales code to throw an exception after some number of events. Ultimately, we intend to make PanScales more tolerant of ill-behaved PDFs, but at this stage of development we have taken the approach that it is safer to abort the run than to continue generating events when a clear problem has arisen, even if only a rare occurrence.

The same directory contains a range of other examples that can be run with Pythia8.3, and the header of each example illustrates how to use it. We also provide an example of interfacing to Rivet [39], in which case one needs to make sure Rivet is installed and the code is compiled with it (through `--cmake-options="-DWITH_RIVET=on"`). This allows for an easy comparison with data, although we should note that none of the PanScales showers are currently tuned and do not include the effects of quark masses.

The PanScales-Pythia8.3 interface transfers the event into the Pythia8.3 event record after each shower emission. This is done to have access to the full Pythia8.3 functionality in the future, i.e. interleaving multi-parton interactions (MPI) with showering of the hard process. Note that at this moment, we do not have the functionality to run with MPI, but hadronisation can be added through the flag `-hadron-level`.

The above examples all use the same PanScales event-loop framework as in the main `shower-code/` directory. We also distribute examples with a standard Pythia8.3 structure. These are to be found in the `main-pythia02.cc` and `main-pythia06.cc` files, which are based on the corresponding `main02.cc` and `main06.cc` examples from the Pythia8.3 distribution. Note that only a restricted set of options is supported in this form, which can be found at the end of `PanScalesPythiaModule.cc`.

## 2.5   Code validation and more advanced builds

To validate that the code is generating expected results, enter the `shower-code/validation/` directory and run

```
./validate-showers.py -j
```

which runs a range of validation tests in parallel across all available cores. This takes a total of about $300-1000$ CPU seconds on a typical laptop, carrying out of the order of 100 separate short runs, each with different settings, verifying that they give histograms that are identical to those in a set of reference files. It is possible to carry out validation runs with larger numbers of events, but one should be aware that there can be differences due to varying floating-point behaviours across different hardware. Additionally, lower-level unit tests can be found in the `unit-tests/` directory. The unit tests and a subset of the validation tests can also be run with the continuous integration script `scripts/CI-build-and-test.py`.

---

[3]The code can also be used without LHAPDF, instead using a replacement toy PDF set. See `pythia-interface/README.md` for details.

It is sometimes useful to build the main code in different numerical precisions, e.g. for logarithmic tests that probe very disparate energy scales and angles. For this, the general build script has an option `--builds X` which essentially invokes `cmake` with a suitable set of configuration options. Specifically, one runs

```
../scripts/build.py --builds X [-j]
```

where `X` is a space-separated list that contains one or more of the following options: `double`, `ddreal`, `qdreal`, `doubleexp`, `mpfr4096`.

The `ddreal` and `qdreal` options require at least version 2.3.23 of the QD library [28] and have precisions of about twice and four times that of a double type, with speeds that are about 10 and $200-300$ times slower.

The `doubleexp` type was developed specifically for the PANSCALES project and has the same relative precision as `double`, but a much larger range of exponents (stored in an additional 64-bit integer), which is useful when exploring finite values of $\alpha_s \ln v$ with very small values of $\alpha_s$ and correspondingly large values of $\ln v$. It is about $3-10$ times slower than `double`.

The `mpfr4096` type is based on the MPFR library and has 4096 bits for the mantissa, i.e. about 75 times higher than double precision, or equivalently a little over 1000 decimal digits of precision. It is about $1500-2500$ times slower than `double` (using version 4.2.1 of the library).

For most purposes, the `double` and `doubleexp` types are sufficient, notably when used with methods that track the differences between directions of dipole ends in addition to the actual momenta of the corresponding partons [2]. That tracking can be enabled at run time with the `-use-diffs` option and has only a modest speed penalty, of the order of 10%. The higher precision types are, however, important when developing new showers and testing the correctness of any parts of the code that carry out the dedicated calculations with direction differences.

## 3   Further details for logarithmic tests

In this section we provide some insight into features of the code that facilitate shower logarithmic accuracy tests, together with a more detailed discussion of some of the underlying methodology than has been given in previous work.

Some of the discussion below concerns tests that go beyond the simple ones illustrated in section 2.3. Code for these more advanced tests is to be found in the directory `analyses/nll-validation/` with usage explanations in the corresponding `README.md` file.

### 3.1   Global event shapes

We start by considering the global event shape tests of section 2.3 and specifically examine the commands that are run by the `example-global-nll-ee.py` script. For each of several $\alpha_s$ values, that script executes one or more commands of the following kind

```
build-double/example-global-nll-ee -Q 1.0 -shower panglobal -beta 0.0 \
      -alphas 0.0016 -nloops 2 \
      -lambda-obs-min -0.5 -lnkt-cutoff -327.5 -dynamic-lncutoff -15 \
      -weighted-generation -nev-cache 75000.0 \
      -spin-corr off -use-diffs -nev 750000 -rseq 11 \
      -out example-results/lambda-0.5-alphas0.0016-rseq11.res
```

The second line indicates the value of $\alpha_s(Q)$ in the $\overline{\text{MS}}$ scheme (working in units of $\sqrt{s} \equiv Q = 1$) and that a two-loop running coupling is to be used. The next two lines contain two separate critical elements, which we discuss below in more detail. Further options are `-spin-corr off`, which turns off spin correlations (leaving them on adds about 50% to the run time). The `-use-diffs` option turns on tracking of direction differences for higher numerical precision. It is not strictly necessary for this example, but the speed cost is relatively limited, at about 10%. The `-rseq 11` argument specifies the random sequence that is used.

The two critical elements that we now need to discuss in more detail are (1) a dynamic generation cutoff, which is necessary to prevent event particle multiplicities from becoming intractable and (2) weighted event generation, which is necessary in order to explore regions with large Sudakov suppression.

### 3.1.1 Dynamic generation cutoff

Let us discuss the following part of the command line:

```
-lambda-obs-min -0.5 -lnkt-cutoff -327.5 -dynamic-lncutoff -15
```

This indicates that the minimal value of $\lambda = \alpha_s \ln v$ is $-0.5$, which corresponds to $\ln k_t/Q = -312.5$ with $\alpha_s = 0.0016$. The choice `-lnkt-cutoff -327.5` allows the shower to run somewhat below the $k_t$ scale associated with the minimal value of $\lambda$. This is important, because multiple emissions break the immediate relation between the shower scale and the observable, e.g. because of shower recoil effects, and because the observable sums contributions from multiple emissions. These effects are properly accounted for only if the shower is allowed to evolve sufficiently far below the single-emission scale that is equivalent to the smallest value of the observable that is probed.

If we were to straightforwardly run with the very small cutoff indicated above, then the average event particle multiplicity, $n$, would be prohibitively large. Specifically, it scales as $\ln n = |\lambda_c|\sqrt{2C_A/(\pi\alpha_s)} + \mathcal{O}(1)$ [40], where $\lambda_c = \alpha_s \ln k_{t,\text{cutoff}}/Q$. For the parameters above, this would give $n \sim 10^8$. To resolve this issue, in addition to the fixed $k_t$ cutoff we also use a dynamic cutoff. We track the shower scale $v_1$ of the first shower emission. Knowing that we are restricting our attention to $\beta_{\text{obs}} = \beta_{\text{PS}}$ observables and that standard global event shapes are recursively infrared-and collinear safe [41], we can be sure that emissions with much lower values of $v$ will not modify the event shapes. Accordingly, when the shower reaches an $\ln v$ scale that is sufficiently far below that of the first emission, we simply stop showering. That choice is specified by the `-dynamic-lncutoff -15` argument, which stops the showering when $\ln v < \ln v_1 + \delta_{\ln v} = \ln v_1 - 15$. This ensures that the multiplicity is kept limited, e.g. for the above run it averages to about 20.

To validate this approach, the dependence of the normalised cross section $\Sigma(\lambda)$ on the size of the dynamic cutoff is shown in Fig. 2a (red line, corresponding to a $k_t$-like observable, i.e. $\beta_{\text{obs}} = 0$). It is illustrated for the $pp \to Z$ process, as carried out for the studies of Ref. [6], and one observes good convergence of $\Sigma(\lambda)$ as the dynamic cutoff $\delta_{\ln v}$ is taken below $-9$ (for the $\beta_{\text{PS}} = \beta_{\text{obs}}$ case discussed, $\delta_{\ln v} \equiv \delta_{\ln O}$ in the plot). The value of $-15$ that we use above leaves a comfortable margin.

The above procedure is sufficient when the angular power $\beta_{\text{PS}}$ in the definition of the shower ordering variable coincides with the angular power $\beta_{\text{obs}}$ in the observable. When $\beta_{\text{PS}} \neq \beta_{\text{obs}}$ a more elaborate procedure is needed. As a starting point, we need to be able to evaluate the order of magnitude of the contribution of each emission $i$ to the observable, $O_{\text{approx},i}$. Given knowledge of $\beta_{\text{PS}}$ and $\beta_{\text{obs}}$, this can be determined from the value of

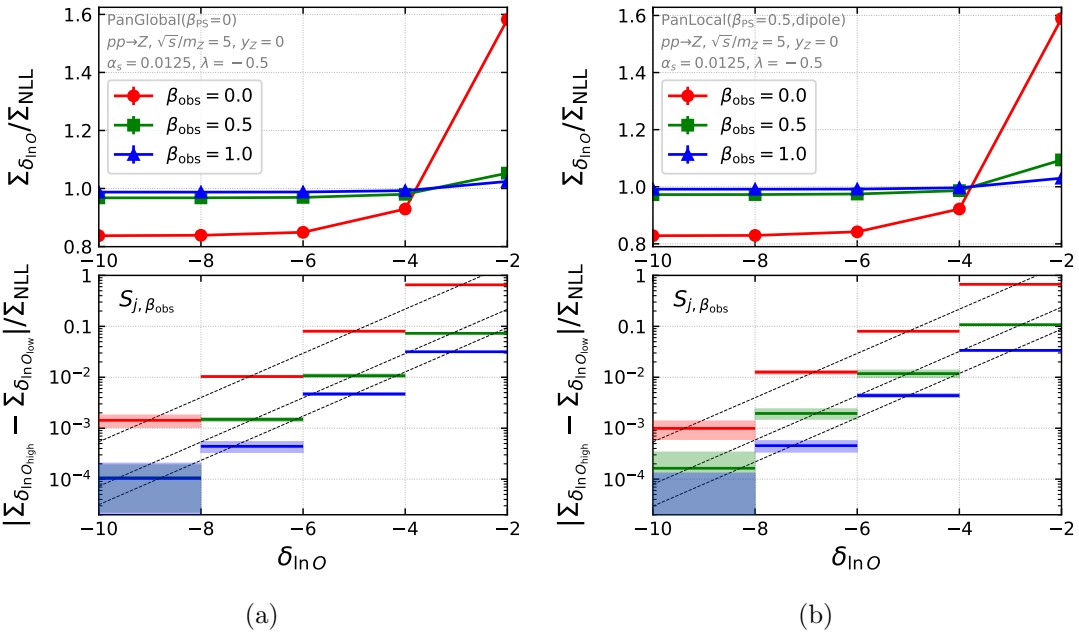

Figure 2: Dependence of $\Sigma(\lambda)$ on the size, $\delta_{\ln O}$, of the parton-shower dynamic cutoff. The results are shown for the $S_{j,\beta_{\mathrm{obs}}}$ class of observables in Drell-Yan production, which sum $p_{ti}e^{-\beta_{\mathrm{obs}}|\eta_i|}$ over all final-state jets $i$ (with $p_t$ defined as the transverse momentum with respect to the beam). The dependence on the size of the dynamic cutoff size is shown normalised to the NLL prediction for $\Sigma$, for each of three values of $\beta_{\mathrm{obs}}$: 0 (red), 0.5 (green) and 1.0 (blue). Two showers are shown (a) PanGlobal with $\beta_{\mathrm{PS}} = 0$ and (b) PanLocal (dipole) with $\beta_{\mathrm{PS}} = 0.5$. The upper panels show the ratio of $\Sigma$ to the NLL result, showing the convergence as $\delta_{\ln O}$ is made more negative (note that the ratio is not expected to go to 1, because of the finite value of $\alpha_s$). The lower panels show the difference between $\Sigma/\Sigma_{\mathrm{NLL}}$ results for successive $\delta_{\ln O}$ values, again normalised to the NLL result, giving a clearer view of the degree of convergence. The dashed lines help illustrate that the behaviour is consistent with exponential dependence on $\delta_{\ln O}$.

$\ln v$ and the (approximate) rapidity of the emission.[4] We then determine if the following condition holds

$$\ln O_{\text{approx},i} < \max_{j<i}\{\ln O_{\text{approx},j}\} + \delta_{\ln O} \tag{1}$$

where on the right-hand side the max operation runs over prior accepted emissions $j$ and $\delta_{\ln O}$ generalises the $\delta_{\ln v}$ discussed above. If Eq. (1) holds, then emission $i$ is discarded and showering continues. Showering is subsequently stopped when $\ln v$ is sufficiently small such that for all emission rapidities one can be sure that Eq. (1) will always hold.

Note that for $\beta_{\text{PS}} \neq \beta_{\text{obs}}$ this procedure is guaranteed to be safe only for showers that respect the PANSCALES condition that a given emission does not impact other emissions far in the Lund plane.[5] Results are illustrated in Fig. 2, for various combinations of $\beta_{\text{PS}}$ and $\beta_{\text{obs}}$, showing the relative change in $\Sigma$ between successive pairs of values of $\delta_{\ln O}$, corresponding to the extremities of each horizontal bar. One sees a behaviour that is consistent with an exponentially vanishing effect as $\delta_{\ln O}$ becomes more negative. Again, a choice of $-15$ should be more than adequate for NLL logarithmic tests. Note that the need for a more sophisticated dynamic veto and cutoff is not the only challenge that arises with $\beta_{\text{PS}} \neq \beta_{\text{obs}}$. Further considerations are discussed below in section 3.1.3.

### 3.1.2 Weighted event generation for $\beta_{\text{ps}} = \beta_{\text{obs}}$

Now we turn to the following part of the command line at the beginning of section 3.1

```
-weighted-generation -nev-cache 75000.0
```

This is needed to address the fact that $\Sigma$ becomes infinitesimally small in the limit $\alpha_s \to 0$ for fixed $\lambda = \alpha_s \ln v$, specifically $\ln \Sigma \sim \lambda/\alpha_s$. To get a more concrete sense of the challenge, consider that $\Sigma \sim 10^{-103}$ for $\lambda = -0.5$ and $\alpha_s = 0.0016$ as in the command line at the start of section 3.1. With unweighted event generation, it would take orders of magnitude longer than the age of the universe to explore that region.

We address this challenge by greatly enhancing the number of events whose first emission has an extremely small value of $\ln v$, assigning a suitable weight to those events so as to reproduce the correct final $\Sigma$ distribution.

We divide the full evolution range $[\ln v_{\text{max}}, \ln v_{\text{min}}]$ into a set of $n$ consecutive bins, each defined by their upper boundaries, $\ln v_i^+$. Writing the shower Sudakov form factor between two scales $v$ and $v'$ as $\Delta(v, v')$ (for the Born event), we precompute the part of the Sudakov form factor associated with each bin.[6] The precomputed Sudakov can be inspected in the output file.

For each event, we choose a generation bin $i$ randomly, with a probability $p_i$ that we take proportional to $\ln v_i^+ - \ln v_{i+1}^+$. The shower then starts from scale $v_i^+$. If the first emission scale is above $v_{i+1}^+$, the shower continues down to the dynamic cutoff as explained in section 3.1.1. If the first emission scale is below $v_{i+1}^+$, the emission is discarded and showering is restarted from scale $v_i^+$, repeating this until one generates an emission above $v_{i+1}^+$. The weight assigned to the event is

$$w = \frac{1}{p_i}\left(\Delta(v_{\text{max}}, v_i^+) - \Delta(v_{\text{max}}, v_{i+1}^+)\right). \tag{2}$$

---

[4]With the help of the `ShowerBase::Element::lnobs_approx(...)` function in the `panscales` namespace.

[5]This condition is not satisfied for standard dipole showers. For $\beta_{\text{obs}} \neq \beta_{\text{PS}}$ this results in super-leading logarithmic terms [1] and such terms would not be fully reproduced with the above dynamic veto procedure.

[6]This can be done either with a Monte Carlo integration (the default) or with Gaussian quadrature. The `-nev-cache` argument indicates the number of events used for the MC integration in each bin. We have found that a suitable choice is about 10% of the total number of events that one wishes to generate.

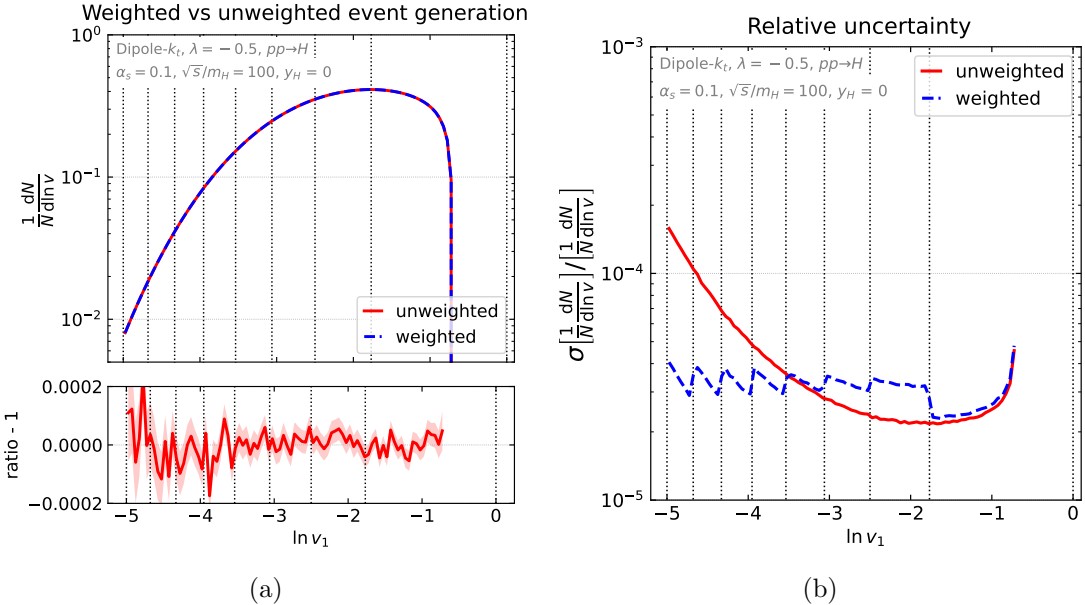

Figure 3: Validation of the weighted event generation. We show the results for Dipole-$k_t$ for $pp \rightarrow H$ collisions with $\sqrt{s}/m_H = 100$ and $y_H = 0$. We take $\alpha_s = 0.1$ and target $\lambda = -0.5$. (a) The distribution of $\ln v_1$ values for the first emission, with and without weighted generation, illustrating that they agree. The bottom panel shows the ratio between the weighted and unweighted results minus 1, where the statistical uncertainty is indicated with the red band. (b) The relative statistical uncertainty in the $\ln v_1$ distribution for both unweighted and weighted shower generation.

where the factor $1/p_i$ compensates for the likelihood of choosing the window and the second factor accounts for the Sudakov form-factor probability of having the first emission between $v_i^+$ and $v_{i+1}^+$.

In practice we choose the bins such that $\ln v_i^+$ scales as $-\sqrt{i}$, which in a fixed-coupling approximation ensures that the Sudakov form factor decreases by a similar factor from one bin to the next. We take the total number of bins to be $n = \ln \Delta(v^+, v^-)/\ln(u)$ where $u$ is a number of $\mathcal{O}(1/2)$, so that the probability of each successive bin is about half that of the previous bin. This ensures that each attempt at starting the showering has a $\sim 50\%$ chance of generating an emission within the given window, while also ensuring that the event weight tracks the actual first-emission Sudakov probability to within about a factor of two.

Fig. 3a shows a validation of the correctness of the procedure for a physical value of the coupling, $\alpha_s = 0.1$, where it is straightforward to obtain high accuracy with both weighted and unweighted approaches. The plot shows the distribution of $\ln v_1$ for the first emission when the generation is unweighted (red solid lines) and weighted (blue dashed lines). The two approaches agree to within statistical errors. Fig. 3b shows the size of the relative statistical error in the two approaches, illustrating the superiority of weighted generation for small $\ln v_1$ values, and also illustrating that its statistical error is fairly independent of $\ln v_1$, with a mild sawtooth structure that lines up with the generation bin edges (indicated as vertical dotted lines).

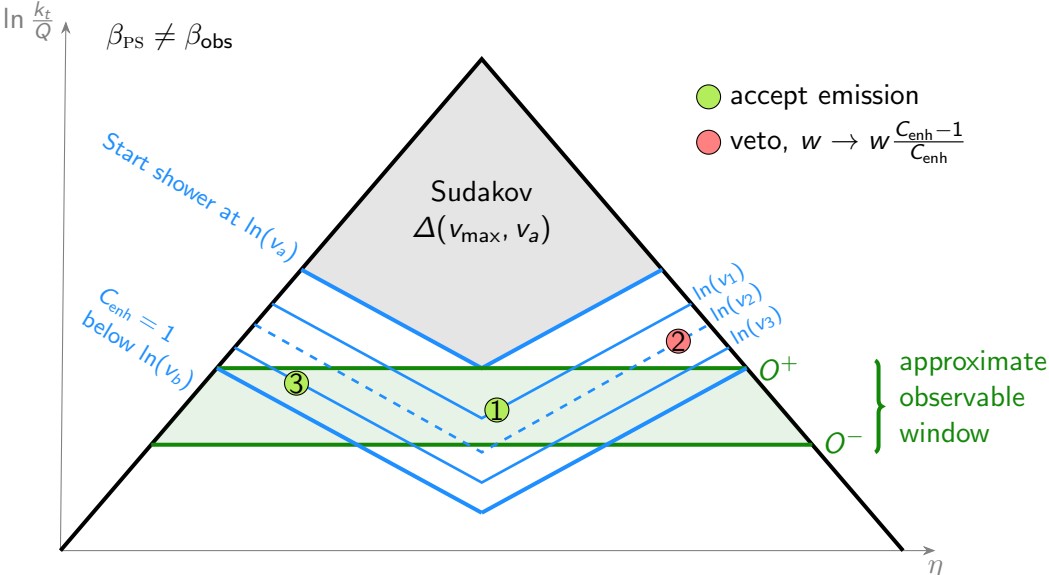

Figure 4: Illustration of some of the main steps in the weighted event-generation approach that is used for $\beta_{\mathrm{PS}} \neq \beta_{\mathrm{obs}}$, for a specific target observable window. Beyond what is shown in the figure, one important element is that if, after showering, there are no emissions in the approximate observable window, the event is discarded. The approach is additionally supplemented with the dynamic generation cutoff of section 3.1.1.

### 3.1.3    Weighted event generation for $\beta_{\mathrm{ps}} \neq \beta_{\mathrm{obs}}$

The above weighted event-generation procedure is very powerful when the Lund contour of the observable lines up with that of the shower evolution variable, i.e. $\beta_{\mathrm{obs}} = \beta_{\mathrm{PS}}$. If this is not the case, a given value of the observable receives contributions from different evolution windows with often widely differing weights, significantly worsening statistical convergence. In these situations, we employ a different procedure, which relates to an approach that was first introduced in Ref. [42] in the context of multi-jet merging. The reader may wish to consult Fig. 4 as they follow the description of the approach. As with the dynamic cutoff in section 3.1.1, it is useful to introduce an approximate (soft-collinear) calculation of the contribution to the observable from any given single emission, $O_{\mathrm{approx},i}$. In a first instance the question is how to efficiently generate events such that $O_{\max} \equiv \max_i\{O_{\mathrm{approx},i}\}$ is in some range $O^- < O_{\max} < O^+$, the green band in Fig. 4, labelled "approximate observable window". Knowing the scaling of the observable, it is possible to analytically work out the largest shower scale, $v_a$, that can generate an approximate emission with observable-value $O^+$. Showering then always starts from that scale $v_a$, with an initial weight equal to $w = \Delta(v_{\max}, v_a)$, where $v_{\max}$ is the largest kinematically accessible scale. We then need to ensure that the showering does not generate any emissions with $O_{\mathrm{approx},i} > O^+$. The simplest approach would be to veto every event that has any of the emissions $i$ contributing such that $O_{\mathrm{approx},i} > O^+$. This would in general lead to a very small fraction of surviving events. Instead, and perhaps somewhat counter-intuitively, we use an enhancement $C_{\mathrm{enh}} > 1$ for the probability of generating individual emissions. If an emission $i$ has $O_{\mathrm{approx},i} > O^+$, the emission is discarded, the event is kept, but the event weight $w$ is multiplied by a factor such that

$$w \to w \, \frac{C_{\mathrm{enh}} - 1}{C_{\mathrm{enh}}}. \tag{3}$$

Emissions with $O_{\text{approx},i} \leq O^+$ (including those with $O_{\text{approx},i} \leq O^-$) are instead accepted with probability $1/C_{\text{enh}}$, with the event weight unchanged. Once $v$ goes below some value $v_b$ such that no further emission can have $O_{\text{approx},i} > O^+$, the enhancement factor is set equal to 1 and the shower continues down to the scale of the dynamic cutoff. After statistical averaging this gives exactly the same result as a uniform-weight approach that discards every event with any emissions $O_{\text{approx},i} > O^+$ [42].[7] Finally, events are only accepted if at least one emission has an approximate observable value in the range $O^- < O_{\text{max}} < O^+$.

The width of the event weight distribution gets smaller as $C_{\text{enh}}$ is made larger, but there is an associated slow-down of the showering, because of the increased emission probability. The optimal choice for $C_{\text{enh}}$ involves some balance between these two aspects. In practice we use $C_{\text{enh}} = 20$ for showers with $\beta_{\text{PS}} = 0$, and $C_{\text{enh}} = 100$ for showers with $\beta_{\text{PS}} = 0.5$.

The final step is that for a given value of the constraint on the actual observable (e.g. $\alpha_s \ln O < \lambda = -0.5$), we need to add together several contiguous $O^{\pm}$ windows above the constraint, plus one final window without a lower bound (i.e. $O^- = 0$). The need for several windows is because the actual value of the observable can be smaller than the approximate value of the observable by some $\mathcal{O}(1)$ factor.

In its current form, the code (`analyses/nll-validation/shower-global-obs.cc`) performs separate runs for each of several windows and the python script that manages the calculations (`run-nll-tests.py`) adds the results together. A command line that sets up the use of the above algorithm is, for instance

```
build-double/shower-global-obs -Q 1.0 -no-spin-corr  -nloops 2 -alphas 0.008 \
      -shower panglobal -beta 0.0 -colour NODS \
      -lambda-obs -0.5 -beta-obs 0.5 -lnkt-cutoff -77.5 \
      -strategy RouletteEnhanced -enhancement-factor 20.0 \
      -ln-obs-buffer 3.5 -nln-obs-div 7 -veto-buffer -15.0 \
      -use-diffs  -rseq 11 -nev 70000 \
      -out nll-tests-tmp/panglobal00-fapx0.5-lambda0.5-as0.008-NODS-rseq11.dat
```

The second and third lines set up a shower and an observable class with different $\beta_{\text{PS}}$ and $\beta_{\text{obs}}$ values. The fourth line sets up the strategy described above and the associated value of the enhancement factor, corresponding to $C_{\text{enh}}$. The fifth line indicates that 7 approximate observable windows are explored, extending to $e^{3.5}$ times above the target value of the observable. The code performs runs in each of the 7 separate approximate-observable windows, producing one output file for each (Fig. 4 corresponds to a single window). The `-veto-buffer -15` argument on the fifth line sets the size of the dynamic veto and cutoff, as explained for $\beta_{\text{PS}} \neq \beta_{\text{obs}}$ at the end of section 3.1.1. If a user wishes to explore new observables or new showers, they are strongly advised to manually verify that the contribution to the cross section from the highest window is sufficiently suppressed relative to the total result. This can be done by examining the output files from the above command, there being one file per observable window.

---

[7]One way of understanding this is to think that one starts with a number of replicas of the shower that will evolve in parallel. The number of replicas is equal to $C_{\text{enh}}$, and the enhancement of the emission probability can be interpreted as being equivalent to the total probability of emissions occurring in any of the replicas. Concentrating specifically on emissions with $O_{\text{approx},i} > O^+$, the first time any of the replicas generates an emission with $O_{\text{approx},i} > O^+$, that replica is simply discarded, leaving $C_{\text{enh}} - 1$ replicas. The factor $\frac{C_{\text{enh}}-1}{C_{\text{enh}}}$ in Eq. (3) is simply the ratio of surviving to original replicas. As the shower continues, the remaining weight $\frac{C_{\text{enh}}-1}{C_{\text{enh}}}$ is then shared back out across $C_{\text{enh}}$ replicas again, and so the procedure continues.

## 3.2 Single-logarithmic observables, e.g. non-global logarithms

An example of a single-logarithmic test, such as the transverse energy flow in a rapidity slice in $e^+e^- \to q\bar{q}$ collisions, can be performed with the example-nonglobal-nll-ee.py script in the shower-code/ directory. This will execute the following command

```
build-doubleexp/example-nonglobal-nll-ee -Q 1.0 -shower panglobal -beta 0.0 \
    -nloops 2 -colour CATwoCF  \
    -slice-maxrap 1.0 -lambda-obs-min -0.5 \
    -alphas 1e-09 -lnkt-cutoff -501000000.0 -ln-obs-margin -11 \
    -strategy CentralRap -half-central-rap-window 10  \
    -spin-corr off -nev 10000 -rseq 11 \
    -out example-results/lambda-0.5-alphas1e-09-rseq11.res
```

The second line indicates that a 2-loop running coupling is to be used and that the colour scheme is a large-$N_{\rm C}$ scheme in which $C_A = 2C_F = 8/3$.[8] The third line indicates the size of the rapidity slice in which the in-slice energy flow $E_t$ will be measured, and sets the minimum value of $\lambda = \alpha_s \ln E_t/Q = -0.5$.

The fourth line indicates that we run at an infinitesimal value of the coupling, $\alpha_s(Q) = 10^{-9}$ (still working in units of $\sqrt{s} \equiv Q = 1$), with a sufficiently small $k_t$ cutoff, $e^{-501000000}$, so as to cover the region down to $\lambda = -0.5$. The point of using such a small value of $\alpha_s$ is to avoid a substantial extrapolation to $\alpha_s = 0$, alleviating the need for runs at multiple values of $\alpha_s$. It is physically possible to use such an infinitesimal value because the observable $\Sigma(\lambda, \alpha_s)$ is independent of $\alpha_s$ for $\alpha_s \to 0$. This is in contrast with the case of global event shapes where $\ln \Sigma(\lambda, \alpha_s) \sim \lambda/\alpha_s$. The huge range of orders of magnitude of momenta requires the use of the doubleexp type.

As with the case of global event-shape tests, a straightforward run with the above parameters would give multiplicities that are much too high to be managed (in fact, the situation is even worse, because of the much smaller value of $\alpha_s$). The mitigation procedure is different in this case: the combination of arguments on the fifth line causes the shower to only generate emissions within a window where the absolute rapidity is less than 11 (with respect to the emitter or spectator). For showers that satisfy the PANSCALES conditions, as long as $\lambda$ is not too large and the window is large enough, the results should be (and are) independent of the window size. Corresponding arguments also exist for an analogous modification of the shower to generate only hard-collinear emissions, as is relevant for many spin-correlation, fragmentation function and PDF evolution tests.

Note that for verifying next-to-single-logarithmic accuracy [12, 13, 45] for non-global logarithms [9] it is necessary to enable the double-soft matrix element corrections and associated virtual corrections (with the -double-soft command-line argument, available only for the PanGlobal showers with $\beta = 0$ and $\beta = 0.5$, for $e^+e^-$ collisions). One also has to modify the above command line to run at multiple small but finite values of $\alpha_s$, so as to be able to determine the first derivative of $\Sigma(\lambda, \alpha_s)$ with respect to $\alpha_s$ in the $\alpha_s \to 0$ limit. Doing so accurately requires considerable computer resources. Users who wish to explore this are advised, in a first instance, to run with the -split-dipole-frame option for the PanGlobal showers with $\beta_{\rm PS} = 0$, which is the most computationally efficient setup.

---

[8] Recall that the PanScales showers are logarithmically accurate for non-global logarithms only in the large-$N_{\rm C}$ limit, though the subleading-colour schemes [2] are numerically close to the full-colour results [43, 44]. The schemes of Ref. [2] can be obtained by replacing CATwoCF with NODS or Segment.

# 4 Implementing a new shower

The PanScales framework allows for relatively straightforward addition of new dipole and antenna showers with alternative kinematic maps or ordering variables. This allows the user to leverage the existing code for colour and spin handling, as well as the infrastructure for logarithmic accuracy tests and the interface to Pythia8.3. As an example, our own toy implementation of the Pythia8.3 final-state shower involves about 250 lines of header (`ShowerToyPythia8.hh`), most of which are boilerplate code, and a further 250 lines in `ShowerToyPythia8.cc`, more than half of which are comments or blank lines. For a slightly more elaborate example that handles also initial-state radiation, the user may wish to look at the `ShowerDipoleKt` class.

Here we outline some of the aspects that a user should keep in mind in implementing their own new shower. Firstly, the code is in the `panscales` namespace. Inspecting the code, the user will see that rather than `double`, many variables are in `precision_type`: this corresponds to the precision that was chosen in the build step. Generically, `double` is used for logarithmic variables and acceptance probabilities, while `precision_type` is to be used for (non-logarithmic) kinematic variables and associated matrix-element calculations, where rounding errors and large exponents may be encountered.

Much of the core work of showering is carried out by the `ShowerRunner` class, which is essentially agnostic as to the specific details of any given shower. The basic work flow of `ShowerRunner` is depicted in Fig. 5. The `ShowerRunner` class is constructed by passing a pointer to any class that derives from `ShowerBase`, whose role is to handle a small, well-defined set of shower-specific tasks, such as providing the acceptance probability and implementing the shower's kinematic map. The functions that the user should provide to the shower are also shown in Fig. 5. For concreteness, with this distribution, we have supplied two files, `ShowerUserDefined.hh` and `ShowerUserDefined.cc`, derived from `ShowerBase`, which one can modify. The functions that need to be touched are marked with "USER-TODO" in the code. The user-defined shower can then be run with the `-shower user-defined` command line argument. Note that the template is meant for $e^+e^-$ showers. For additional features like initial-state radiation, matching or the implementation of double-soft currents, the reader should examine the structure of other existing showers.

The class `ShowerUserDefined` implements:

- Member functions that provide textual descriptive information about the shower.

- Member functions that provide structural information about the shower, notably whether the shower is a dipole shower (only the emitter splits), or an antenna shower (both the emitter and spectator can split) via the function `only_emitter_splits()`, and `n_elements_per_dipoles()` (two for a dipole shower, one for an antenna). For showers with a global kinematic map, one where a dipole branching also impacts the kinematics of particles not in the dipole, the function `is_global()` should return `true` (more on this later).

- Member functions to create the two sub-classes `ShowerUserDefined::EmissionInfo`, and `ShowerUserDefined::Element`.

The two sub-classes encode the data associated with an emission and most of the implementation of the shower splitting:

- `ShowerUserDefined::EmissionInfo` derives from `ShowerBase::EmissionInfo`. A pointer to its base class is used by `ShowerRunner` as the main structure to keep

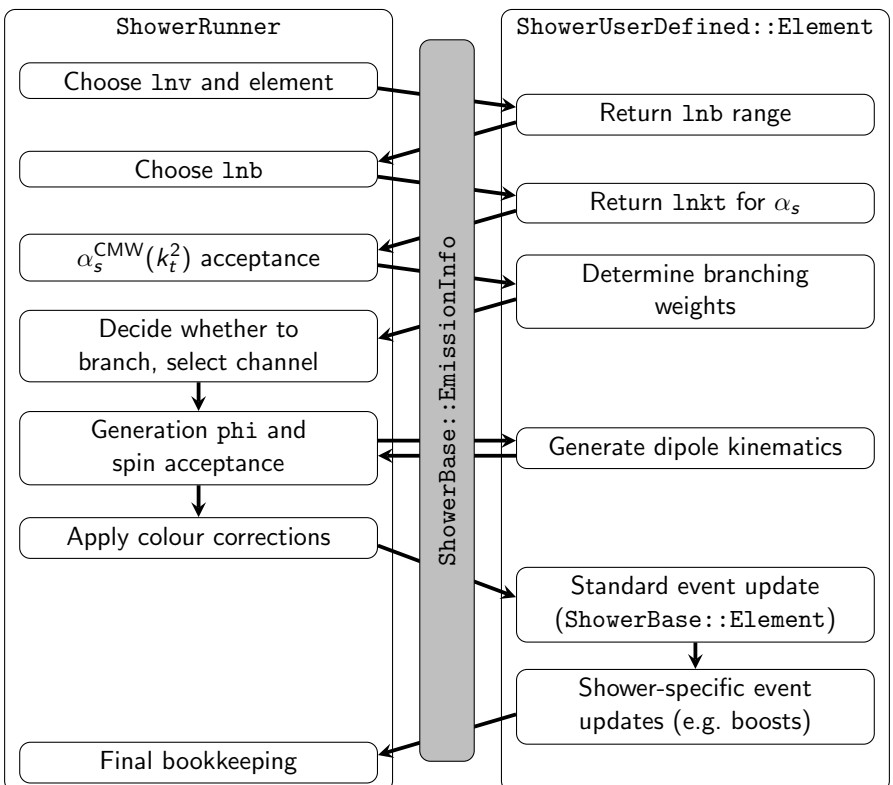

Figure 5: Simplified structure of a single trial for an emission in the shower. Steps on the left are performed by the centralised `ShowerRunner` class. Shower-specific steps, shown on the right, are the responsibility of the `ShowerUserDefined` implementation, as well as its sub-classes. The `EmissionInfo` sub-class stores all required information needed for the current branching, such as the splitting variables and constructed kinematics. The emitting dipole is accessible in the `Element` sub-class, which contains the majority of the shower-specific code. The matching, implementation of the spin correlations throughout the event evolution, and the double-soft splitting functionalities are not illustrated here.

track of the emission through the various steps of constructing that emission, and is generally passed to all shower-specific functions. Much of the required functionality is already present in the base class. Users may extend it to cache quantities computed at specific steps of a single emission that is trialled, so as to not reproduce them again at a later stage. An example would be the absolute $k_t$ of an emission, or the longitudinal momentum fraction $z$, which might first be calculated in the early stages of Fig. 5 and then used later when generating the new dipole kinematics.

- `ShowerUserDefined::Element` derives from `ShowerBase::Element`. It is responsible for carrying out almost all of the shower-specific work. In case of multiple types of elements (i.e. II, IF and FF dipoles), one might choose to derive distinct element classes for each, typically each from an intermediate `ShowerUserDefined::Element` base class (see e.g. `ShowerDipoleKt`).

Let us focus here on the `Element` class. For a dipole shower, where each dipole end is associated with a distinct kinematic map, there will be two `Element`s per dipole, one for each end. For an antenna shower, there will be just one `Element` per dipole. The branching kinematics are governed by three variables:

- $\ln v$, the logarithm of the (dimensionful) ordering variable.

- $\ln b$, a logarithmic variable for the longitudinal degree of freedom of the emission. In the soft-collinear region of a back-to-back dipole it might, for example, map directly to a rapidity, or to $\ln z$ where $z$ is the radiated particle's momentum fraction.

- $\phi$, the azimuthal angle for the emission.

The main non-trivial member functions that need to be implemented are the following:

- `lnb_extent_const()` and `lnb_extent_lnv_coeff()`: for any given $\ln v$, `ShowerRunner` will take the $\ln b$ generation range to be taken equal to

$$\texttt{lnb\_extent\_const()} + \ln v \times \texttt{lnb\_extent\_lnv\_coeff()} \tag{4}$$

where typically the first term would be a function of the dipole kinematics (as encoded in the corresponding `Element`) and the second term would not. The use of a simple analytic form for the extent facilitates the generation of the $\ln v$ distribution.

- `lnb_generation_range(lnv)`: for a given value of the evolution scale `lnv`, returns a `Range` object containing the limits of $\ln b$ generation. The difference between the upper and lower limits should coincide with the extent expected from the previous bullet point. Not all $\ln b$ values in the range need be kinematically valid.

- `lnv_lnb_max_density()`: returns the maximal emission density (for a $d\ln v\, d\ln b\, \frac{d\phi}{2\pi}$ measure). Typically this would be the soft-collinear limit of the emission density

$$\frac{dP}{d\ln v\, d\ln b\, d\phi/2\pi} = \frac{C_A \alpha_s^{\max}}{\pi} \tag{5}$$

where $\alpha_s^{\max}$ is the maximal value that the strong coupling can take.[9]

- `lnkt_approx(lnv,lnb)`: returns the logarithm of the transverse momentum of an emitted parton with that $\ln v$, $\ln b$ combination. The result should be exact in the soft-collinear limit, but does not need to be exact in the soft large-angle or in the hard-collinear limits. It is typically used for the evaluation of $\alpha_s(k_t)$, as well as for some of the dynamic emission vetoing used for logarithmic accuracy tests.

- `eta_approx(lnv,lnb)`: similar, but returns the rapidity of the emitted parton in the soft-collinear limit, used in the computation of colour transition points and dynamic emission vetoing.

- `acceptance_probability(emission_info)`: sets information subsequently used by `ShowerRunner` in order to calculate the probability that the dipole splits. It makes use of the generation variables `lnv` and `lnb`, as cached in `emission_info`. Specifically it sets the following member variables in `emission_info`:

  - `emitter_weight_rad_gluon` : weight for the emitter to radiate a gluon;

  - `emitter_weight_rad_quark` : weight for the emitter to radiate a (anti-)quark;

  - `spectator_weight_rad_gluon` : weight for the spectator to radiate a gluon;

---

[9] All current shower implementations have a private member pointer `Element::_shower`, and the base shower class has a `max_alphas()` member, as well as a `qcd()` member that supply access respectively to $\alpha_s^{\max}$ and QCD constants. Together these provide the information needed to calculate the maximum density.

– `spectator_weight_rad_quark` : weight for the spectator to radiate a (anti-)quark.

The weights to radiate a gluon/quark depend on the splitting functions. Only for an antenna shower will the spectator weights be non-zero. They are to include a factor of the maximal allowed value of the strong coupling $\alpha_s^{\max}$, with the `ShowerRunner` class then accounting for an $\alpha_s(k_t)/\alpha_s^{\max}$ factor. In initial-state branchings, `ShowerRunner` also accounts for an appropriate PDF ratio factor. The user may choose to use the splitting functions implemented in `QCD.hh`. This can be done by calling `fill_dglap_splitting_weights`, which needs the longitudinal radiated momentum fraction $z$ (where $z \to 0$ indicates the soft limit). In addition, when using spin correlations, the user should set

– `z_radiation_wrt_emitter` : collinear momentum fraction with respect to the emitter;

– `z_radiation_wrt_spectator`: collinear momentum fraction with respect to the spectator.

The `acceptance_probability(...)` function returns a `bool`, where `false` indicates that the generation variables are definitely outside the kinematic limit. For most showers, if it returns `true` then the generation variables would normally be inside the kinematic limit. The emitter and spectator splitting probabilities are then used later in `ShowerRunner` to accept/reject a splitting, and choose the splitting channel.

- `do_kinematics(emission_info, rp)`: constructs the post-branching momenta of the emitter, the spectator and the newly radiated particle. For this, one may again use the cached generation variables `lnv`, `lnb` and `phi`, alongside any other quantity that the user stored in `emission_info`. The post-branching momenta should be stored in `emission_info` under the names `emitter_out`, `spectator_out` and `radiation`. This function returns a `bool`, where `true` indicates that the generation variables were inside the physical kinematic limit.[10] Note that the pre-branching emitter and spectator momenta should be taken from a variable `rp` of type `RotatedPieces`. This class is part of the framework for handling directional differences. It provides a rotated version of the dipole with one or other of its particles aligned along the $z$ axis, which allows the user to retain high precision in the branching kinematics (specifically, small components along the $x$ or $y$ axis) without explicit knowledge of the underlying direction-difference structures. The direction-difference infrastructure then takes care of deducing the correct momenta and direction differences in the original frame.

- `update_event(...)`: In the `Element` base class, the `update_event(...)` member function takes care of replacing the pre-branching emitter and spectator particles with the post-branching ones, adds the radiated particle to the event, and takes care of some of the bookkeeping associated with colour handling. However, other particles in the event are by default not modified. Therefore, for showers

---

[10]It is possible for the `acceptance_probability(...)` function to return true even outside the kinematic limits. In this case there are two possible avenues for imposing the kinematic limit. One is for the shower's `Element` class to overload the base-class member function `check_after_acceptance_probability(...)`, which gets called after $\alpha_s$ and PDF factors have been incorporated into the branching probabilities and those have been used to decide to continue with the emission generation. It is called before $\phi$ and the splitting channel are known. Alternatively `do_kinematics(...)` is called with knowledge of the $\phi$ value and channel, and can return `false` if the emission is outside the kinematics. The former is the only route that is currently valid in order for spin correlations to be correctly accounted for.

with a global kinematic map, this function needs to implement additional operations on the rest of the event (e.g. a boost or rescaling). These would typically be preceded by an explicit call to `ShowerBase::Element::update_event(...)`. The event is then further processed by `ShowerRunner`, updating the event dipoles, colour and spin-density structure in addition to any caching associated with the event generation. Note that cached information associated with dipoles other than the newly-created dipole and the splitting dipole are by default not updated, unless the `ShowerUserDefined::is_global()` function returns true.

Once implemented, the new shower can be run by using the flag `-shower user-defined`.

If the user would like their shower to work with direction differences, they should inspect how this is implemented in existing showers. Aside from the `RotatedPieces` discussion above, calculations of dot products in determining dipole invariants should make explicit use of knowledge of direction differences (available from the `dirdiff_3_minus_3bar` member variable of `element.dipole()`), and `do_kinematics(...)` should use 3-vectors in its internal calculations to avoid triggering off-mass-shell errors. Furthermore if the shower carries out any global boosts, these need to be performed in a way that correctly boosts also the full set of dipole direction differences. The user is invited to inspect the code of existing showers for further details.

It is important also to test the correctness of the direction-differences implementation. Typically we do this by first running a double-precision build with a physically sensible range and verifying that results are identical with/without the `-use-diffs` argument. Then we create a build in the `doubleexp` type, running with `-use-diffs` and a logarithmic range of about 1000 (and correspondingly low $\alpha_s$, so that the multiplicity stays of the order of $10-100$) and compare the output to a build with the much slower `mpfr4096` type, running without `-use-diffs`. Again the results should be identical, though typically a few iterations are likely to be necessary to identify all sources of potential loss of precision. A final comment is that, by default, the logarithmic-accuracy tests of section 2.3 run in double precision, with a $\ln v$ range reaching about 300. However, for this to work, the shower should not ever do more than take the square of a momentum, otherwise the result will exceed the exponent that can be represented in double precision. If this is a problem, the user should modify a configuration flag in `example-global-nll-ee.py` so as to use `doubleexp` (which is somewhat slower).

## 5    Conclusions

This 0.1 series release of the PanScales code allows users to start exploring its features and techniques, notably for tests of logarithmic accuracy of parton showers. It also demonstrates an early version of the interface to Pythia8.3. While we do not yet recommend its use for phenomenological applications, we hope that this early release of the code will provide a foundation for exploring connections with other projects, so as to enable the wide ecosystem of collider physics tools to benefit from the validated logarithmic accuracy of the PanScales showers.

## Acknowledgements

We thank Frédéric Dreyer for his contributions to the PanScales project and code during the initial stages of the project. We are grateful to Peter Skands and Silvia Zanoli for testing a pre-release version of the code and for helpful comments.

This work has been funded by the European Research Council (ERC) under the European Union's Horizon 2020 research and innovation program (grant agreement No 788223), by a Royal Society Research Professorship (RP\R1\180112, GPS and LS, and RP\R\231001, GPS) and by the Science and Technology Facilities Council (STFC) under grants ST/T000864/1 (MvB, GPS), ST/X000761/1 (GPS), ST/T000856/1 (KH) and ST/X000516/1 (KH), ST/T001038/1 (MD) and ST/00077X/1 (MD). LS is supported by the Australian Research Council through a Discovery Early Career Researcher Award (project number DE230100867). The work of PM is funded by the European Union (ERC, grant agreement No. 101044599). Views and opinions expressed are however those of the authors only and do not necessarily reflect those of the European Union or the European Research Council Executive Agency. Neither the European Union nor the granting authority can be held responsible for them. We also thank each others' institutes for hospitality during the course of this work.

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
