# Peer review of "Introduction to the PanScales framework, version 0.1"

_SciPost Physics Codebases_

## Round 1 · Referee Report · Anonymous (Referee 1) · 2024-2-23

Strengths

1- The paper is written very clearly 2- Many examples

Weaknesses

1- A bit technical but this is the very nature of scientific software documentation. 2- Interplay of parton shower and hadronization could have been pointed out more explicitly.

Report

The authors report on version 0.1 of the PanScales code for parton shower simulations. The code is primarily intended to study the logarithmic accuracy of various parton shower variants.

Many examples explain the scope of the program package and possible applications. There are pointers for the user to experiment with the program as well as hints towards extending the package to individual use cases. Both are explained in quite some detail such that a potential user will be able to do whatever they intend to.

Pointers to the Pythia 8.3 program package are also explained with the right balance of cautiousness and encouragement for experimentation. The part of cautiousness could be more explicit. A large effort is made to tune parton showers to hadronization models. It is not clear whether such an attempt has been made here. If yes, it should be written clearly, if not, it should be stated clearly that there might be an interplay between parton shower and hadronization model that leads to an alteration of physical observables. Particularly in the logarithmically enhanced region that is addressed by this package there might be some non-negligible overlap.

Apart from these caveats which might be clear for the informed user, the paper is clearly recommended for publication.
  • validity: top
  • significance: high
  • originality: top
  • clarity: top
  • formatting: perfect
  • grammar: perfect

Author:  Melissa van Beekveld  on 2024-04-21  [id 4435]

(in reply to Report 1 on 2024-02-23)
Category:
answer to question

We fully agree with the referee on the importance of the interplay with hadronisation for phenomenology, and notes of caution were already present in several places, as indicated in the following (together with a description of any changes/additions made): - Introduction (penultimate paragraph) stated: "the code is not yet at a stage of maturity that is suitable for extensive comparisons to experimental data. This is notably because of the absence of finite quark-mass effects, the need for further work on matching with higher (fixed) order effects, as well as tuning of the non-perturbative parameters of the shower and of any hadronisation model with which it is used." We have supplemented the sentence with references to some tuning papers. - Section 2.4 (Pythia interface, 3rd paragraph from end) stated: "none of the PanScales showers are currently tuned and do not include the effects of quark masses." This has now been rephrased to read "nor do they include effects of quark masses. We comment further at the end of this subsection." and have added a paragraph at the end of section 2.4 to discuss the question. - The conclusions stated: "we do not yet recommend its use for phenomenological applications". We have left this unchanged.

---

## Round 1 · Referee Report · Anonymous (Referee 3) · 2024-3-4

Strengths

1- This is a comprehensive description of the main features of the PanScales code base 2- Details usage using examples to demonstrate various features

Weaknesses

3- When the interface with Pythia8 is mentioned, it is essential to discuss the interplay of the NLL (or beyond) accurate shower and a non-first-principles based hadronisation. In particular which criteria have to adhered to in choosing/tuning hadronisation parameters as not to degrade the PanScales accuracy 4- The structure of the code base seems not to directly follow the physics description. While this is not a problem as such, it makes it harder for the user to navigate it. Here a top-down description of the directory structure and what to find where would be helpful before describing the examples.

Report

The manuscript is generally well written and suitably concise. In addition to the above weaknesses, I ask the authors to address the following points 1- The integration of Catch2 as a unit testing library is mentioned, but not discussed. If not for the main code, then at least at the end when the authors describe how users could possibly add their own shower evolution, the authors should mention how the existing incorporation of Catch2 could be used to embed unit testing here. A code example/snippet may be useful. 2- The description of the weighted event generation for the case that the observable does not line up with the shower evolution variable is somewhat convoluted and in places hard to follow. It would benefit from a clear structuring. In addition, the authors mention that the observable is consered here only in its soft-collinear approximation, hence, as there may be a difference between O and O_approx, slices above O^+ and below O^- need to be considered, correct? Is it possible that while O_approx,1<O_approx,2, the real O_2\<O_1? If so, what would need to be done in this case? 3- When considering non-global logs, as the authors comment, single-logarithmic terms are independent of alpha_s in the alpha_s->0 limit. Is it guaranteed that the given numerical value of alpha_s is small enough to be in that limit and not find themselves on some plateau which is different from the actual limit? What informs the given choice? 4- Similarly, the authors work in the Nc->infinity limit where CA=2CF holds. Is there a reason to choose to fix CF=4/3 and set CA=8/3 instead of the often used fixing of CA=3 and setting CF=3/2?

  • validity: high
  • significance: good
  • originality: good
  • clarity: good
  • formatting: good
  • grammar: good

Author:  Melissa van Beekveld  on 2024-04-21  [id 4437]

(in reply to Report 3 on 2024-03-04)
Category:
answer to question

Regarding weakness 3, in response also to a similar query from Referee 1, at the end of section 2.4 we have added a paragraph discussing this question. Regarding weakness 4, in response to this point and a similar one from Referee 2, we have added an explicit outline of the directory structure at the start of Section 2.1.

  1. Indeed, we should have mentioned Catch2 in the manuscript. We have now added a link to it in Section 2.5 ("Code validation and more advanced builds"), and also a link to a newly written README.md file in that directory with some brief explanations of how to get started. The unit-testing was mainly used for testing foundational classes during their development, for example the momentum classes (and, in its own directory, the double_exp type). We have not ourselves used it for testing more complex shower implementations and so are hesitant about encouraging users to try it out for that purpose.
  2. We have restructured this section. It now has a second paragraph that outlines the core steps, and each following paragraph attempts to explain a single one of those steps. We have also introduced an equation for the form of $O_{approx,i}$ in the earlier section 3.1.1. We hope that this makes the section clearer. On the "in addition": this is the reason for having multiple slices (not shown in Fig.4). There was some discussion in the text, but we have now extended it (paragraph starting "The final step [...]"). Note that the lowest window has $O^-=0$.
  3. This is not a problem when running at the infinitesimally small values of the coupling used in the single-logarithmic tests for non-global logarithms (we typically take values in the range 1e-6 to 1e-9). Where it could conceivably be a problem is, for example, in extrapolations such as that in Fig. 1, where the values of the coupling are small but not infinitesimal. In this context, the way we think about the problem is that if we extrapolate to zero coupling using an Nth order polynomial, then there is a residual error associated with terms of higher orders. We have added a paragraph about the choice of $\alpha_s$ values in section 3.1 (penultimate paragraph). In general, extrapolation is a delicate procedure, and when exploring new observables or new showers, it is wise to experiment a little in order to assess the overall consistency of the results.
  4. Both are supported. The footnote on colour choices in section 3.2 has been updated to reflect that and to explain why we sometimes prefer to choose $C_A$=8/3 (it pushes the Landau pole further in the infrared for any given value of the coupling)

---

## Round 1 · Referee Report · Anonymous (Referee 2) · 2024-3-4

Strengths

Concise and very explicit in terms of helping the would-be user get started and illustrating main usages.

Weaknesses

Overall, I find the structure of many different subdirectories for different purposes somewhat confusing, and am not sure I would be able to navigate it confidently if I was not explicitly guided as in the paper, but I accept that this may be a personal preference.

Report

The paper is well written and to the point, documenting the first public release of the PanScales codebase. I think it certainly merits publication in this journal but have a few questions/remarks I'd like the authors to address.

  1. The authors say that the PanScales event record only stores the final particles, after the shower, not intermediate stages. I don't have a problem per se with the motivation they give for this, that for global recoils it would require n-squared level storage to keep every stage but have four remarks/questions: First, why not allow an option for the user to choose this anyway, accepting the increased memory usage, if they would like to see and subject the individual showering steps for analysis? Second, what about PanLocal? Third, since the authors mention that their framework can in principle be used to implement other showers, how hard-coded is this restriction? Would other showers also be limited to store only the final partons? And fourth, if the user would like to know the order of emissions according to PanScales, is this order preserved in the order of the final partons (or retrievable in some other way), or not?

Related to these questions, it would perhaps be useful to include a printed example of the PanScales event record, and explain its features.

  1. I tested that all the examples of building and running the code worked on my system. I did not initially have the CT14lo PDF set installed. The first time I ran the ../scripts/build.py --build-lib -j --with-lhapdf, I got an error message: "CMake Warning: Manually-specified variables were not used by the project: WITH_LHAPDF" and the next command failed to execute. I then ran "lhadpdf install CT14lo" and repeated both the first command above (which now did not produce an error) and the example program then ran without issue. It might be helpful to add the instruction to install the CT14lo set before (rather than after) the command. (Implicitly obvious from the commands and from the subsequent discussion, which is why it was a trivial hiccup, but why not make it explicit?)

I also noted that in the Pythia8 examples, it looks (?) like a full branching-by-branching event history is provided, in contrast to the statement the authors made above about the PanScales internal event record. Thus also in this context it was not completely clear to me how strict the restriction on not providing the intermediate steps really is; if it is done anyway (and always?) when interfacing Pythia? And does this depend on whether PanScales drives the evolution or Pythia drives it (like in main-pythia06)?

  • validity: top
  • significance: high
  • originality: high
  • clarity: top
  • formatting: excellent
  • grammar: good

Author:  Melissa van Beekveld  on 2024-04-21  [id 4436]

(in reply to Report 2 on 2024-03-04)
Category:
answer to question

Regarding the subdirectories, we appreciate the referee's concern.
To help with this point, we have added an explicit outline of the
directory structure at the start of Section 2.1.

Regarding the Panscales event record, we have added an example
printout as Appendix A. Furthermore that appendix explains how to
ask PanScales to cache information on the event at each stage of the
shower evolution. We have also added a reference to that Appendix
from section 2.2 (where we have also added a footnote mentioning
that the PanLocal shower has global recoil for initial-state branchings).

Regarding the warning, we agree that this is disturbing and have
modified the build script to eliminate it (PanScales version 0.1.2).
We have also moved the instruction to install the CT14lo set to before
the comment.

In writing the Pythia interface, we structured the underlying
ShowerRunner class so that one could map Pythia-specific calls
(e.g. TimeShower::pTnext and TimeShower::branch) to
individual steps of the PanScales showering in the ShowerRunner class.
As a result, Pythia8 always ends up storing an n^2-sized event record.
This is the case independently of whether PanScales or Pythia drives
the shower evolution.

---

## Editorial Decision

resubmitted